# Bats Are Carriers of Antimicrobial-Resistant *Staphylococcaceae* in Their Skin

**DOI:** 10.3390/antibiotics12020331

**Published:** 2023-02-04

**Authors:** Natália Carrillo Gaeta, João Eduardo Cavalcante Brito, Juliana Maria Nunes Batista, Beatriz Gagete Veríssimo de Mello, Ricardo Augusto Dias, Marcos B. Heinemann

**Affiliations:** 1Laboratory of Bacterial Zoonosis, Department of Preventive Veterinary Medicine and Animal Health, School of Veterinary Medicine and Animal Science, University of São Paulo, São Paulo 05508-270, Brazil; 2Faculdades Integradas Campos Salles, São Paulo 05072-000, Brazil; 3Laboratory of Epidemiology and Biostatistics, Department of Preventive Veterinary Medicine and Animal Health, School of Veterinary Medicine and Animal Science, University of São Paulo, São Paulo 05508-270, Brazil

**Keywords:** One Health, Staphylococci, antimicrobial resistance, surveillance

## Abstract

Bats have emerged as potential carriers of zoonotic viruses and bacteria, including antimicrobial-resistant bacteria. *Staphylococcaceae* has been isolated from their gut and nasopharynx, but there is little information about *Staphylococcaceae* on bat skin. Therefore, this study aimed to decipher the *Staphylococci* species in bat skin and their antimicrobial susceptibility profile. One hundred and forty-seven skin swabs were collected from bats during the spring and summer of 2021 and 2022. Bats were captured in different areas of the Metropolitan Region of São Paulo, Brazil, according to the degree of anthropization: Area 1 (Forested), Area 2 (Rural), Area 3 (Residential-A), Area 4 (Slum-– up to two floors), Area 5 (Residential-B—condo buildings), and Area 6 (Industrial). Swabs were kept in peptone water broth at 37 °C for 12 h when bacterial growth was streaked in Mannitol salt agar and incubated at 37 °C for 24 h. The disc-diffusion test evaluated antimicrobial susceptibility. *Staphylococcaceae* were isolated from 42.8% of bats, mostly from young, from the rural area, and during summer. *M. sciuri* was the most frequent species; *S. aureus* was also isolated. About 95% of isolates were resistant to at least one drug, and most strains were penicillin resistant. Eight isolates were methicillin resistant, and the *mecA* gene was detected in one isolate (*S. haemolyticus*). Antimicrobial resistance is a One Health issue that is not evaluated enough in bats. The results indicate that bats are carriers of clinically meaningful *S. aureus* and antimicrobial-resistant bacteria. Finally, the results suggest that we should intensify action plans to control the spread of resistant bacteria.

## 1. Introduction

Bats are significant pest controllers, plant pollinators, and seed dispensers, which are essential in maintaining the ecological balance [1]. Their loss leads to high economic losses, particularly in agriculture, as bats are predators of many crops and forest pests [2]. However, bats have recently stood out as potential carriers of zoonotic viruses, such as the Ebola virus [3] or the coronavirus [4], and zoonotic bacteria, such as *Bartonella* spp. [5], *Leptospira* spp. [6,7,8], *Borrelia* spp. [9], *Brucella* spp. [10], *Yersinia pestis*, and *Mycobacterium tuberculosis* [11].

The *Staphylococcaceae* is a family of Gram-positive bacteria, including the significant *Staphylococcus* genus. *Staphylococcus* spp. is a Gram-positive cocci genus whose species may be members of the normal gut and skin microbiota. Some *Staphylococcus* (e.g., *S. epidermidis*) play an essential role as the first defense against pathogenic bacteria in a mutualistic relationship with the host [12]. However, few species are either opportunistic or pathogenic/zoonotic. For instance, *S. aureus* colonizes humans and other animals, and most of them show no symptoms. However, *S. aureus* can become pathogenic after breaking physical and immunological barriers, leading to mild to severe infections [13,14]. *Staphylococcus* spp. has been isolated from the gut and nasopharynx of bats [15,16], but little information about them on bat skin is available. *S. aureus* was isolated from the skin of a free-ranging population of the fruit bat *Pteropus livingstonii* in the Comoros Republic [17]. In addition, the skin microbiota of bat species in China was evaluated by 16S rRNA gene sequencing and revealed the presence of *Staphylococcus* spp. [18]. One of the most isolated *Staphylococcaceae* species from animals is the formerly called *Staphylococcus sciuri*, which was reclassified in 2020 into the novel genus *Mammaliicoccus*, and now is named *Mammaliicoccus sciuri* [19]. 

Antimicrobial resistance is a global threat that caused approximately 1.2 million deaths in 2019 [20], and if no action is taken, 10 million deaths are estimated by 2050 [21]. It is widespread in humans, the environment, and animals and is inserted in the One Health context [22]. Antimicrobial-resistant bacteria have been isolated in free-living animals, including bats. They are carriers of resistant *Escherichia coli* [23] and other clinically significant bacteria, such as those resistant to the broadly used antibiotics ampicillin, amoxicillin, and amoxicillin-clavulanate [24]. 

Methicillin-resistant *Staphylococcaceae* (MRS), particularly methicillin-resistant *S. aureus* (MRSA), are responsible for hospital outbreaks throughout the world [25]. It is carried by food and companion and wild animals [26]. These bacteria are resistant to beta-lactams, conferred by the *mecA* gene and, less often, the *mecC* gene [27]. The *mecA* is acquired by the mobile genetic element *SCCmec* (staphylococcal cassette chromosome *mec*) [28] and encodes the Penicillin-Binding Protein 2a (PBP2a), which is involved in bacterial cell wall synthesis. The low affinity of this enzyme to β-lactams results in bacterial resistance to all drugs of this antibiotic class [29]. The genetic complexity of the *Staphylococcaceae* became more apparent with the discovery of plasmid-borne methicillin resistance in this family [30]. Methicillin-resistant *Staphylococcaceae* has been detected in wildlife, including bats. For example, in 2007, these bacteria, particularly the *M. sciuri,* were obtained from bat guano in Poland. A year later, MRS was found in the wound and gastrointestinal tract of bats from Germany [31], and recently low resistance rates and the absence of MRSA were observed in bats in the United Kingdom [32].

The synanthropization and the ability of bats to fly to different regions make them significant carriers of antimicrobial-resistant bacteria, including *Staphylococcaceae*, which colonizes their skin and that of several animals [16,17]. Thus, this study aimed to evaluate the species and the antimicrobial susceptibility profile of *Staphylococcaceae* from the skin of bats in São Paulo city, Brazil.

## 2. Results

### 2.1. Number of Bats Captured and Sampled

A hundred and forty-seven swabs from males (N = 75) and females (N = 72), young (N = 27) and adults (N = 120) bats were sampled for this study. Samples were mainly obtained in the rural area (area 2; N = 61), followed by forested area (area 1; N = 40), residential-B (area 05; N = 19), residential-A (area 3; N = 14), slum (area 04; N = 08), and industrial (area 06; N = 07). Nine bat species were captured and sampled: *Sturnira lilium* (N = 56)*, Artibeus lituratus* (N = 38)*, Artibeus fimbriatus* (N = 28)*, Glossophaga soricina* (N = 11), *Carollia perspicillata* (N = 4), *Platyrrhinus lineatus* (N = 4), *Platyrrhinus recifinus* (N = 02), *Pygoderma bilabiatum* (N = 01), and *Desmodus rotundus* (N = 01) all belonging to the family *Phyllostomidae* (Table 1).

### 2.2. Staphylococcaceae Richness

*Staphylococcaceae* were isolated from 42.8% (63/147) of total bats (one isolate per bat), 44.0% (33/75) from males, and 41.7% (30/72) from females (*P* = 0.46), and mostly from young bats (77.8%; 21/27) compared with adult bats (35.0%; 42/120) (*P* = 0.0001). Regarding the study location, most isolates were obtained from area 2 (57.4 %; 35/61), followed by area 1 (45.0 %; 18/40), area 04 (33.3%; 04/12), area 03 (25.0 %; 02/08), area 05 (15.8 %; 03/19), and area 06 (14.3%; 01/07) (*P* = 0.01). More isolates were obtained in the summer (34,0 %; 50/147) than in the autumn (8.8%; 13/147) (*P* = 0.008).

The 63 isolates belonged to 10 species of *Staphylococcaceae. M. sciuri* was the most frequent species, isolated in 66.7% (06/09) of bat species (Table 1)*,* and the *Staphylococcus* isolated were *S. aureus, S. saprophyticus, S. warneri, S. xylosus, S. kloosii, S. epidermidis, S. haemolyticus,* and *S. nepalensis* (Figure 1). The *S. aureus* belonged to spa type t645 (N = 05) or t1451 (N = 02).

Finally, 33 isolates were Gram-negative bacteria whose results will not be described here. No growth was observed in 51 samples.

### 2.3. Diversity Analysis

The diversity was numerically higher in *Artibeus lituratus,* followed by *Glossophaga soricina, Artibeus fimbriatus, Platyrrhinus lineatus,* and *Sturnira lilium* (Figure 2A), but the Shannon index analysis revealed no differences between bat species and *Staphylococcaceae*. (*P* = 0.43). The same analysis was conducted for *Staphylococcaceae* and zone and showed no differences (*P* = 0.41) (Figure 2B).

### 2.4. Antimicrobial Susceptibility Profile

Regarding antibiotic susceptibility, 95.2% (40/63) were resistant to at least one drug tested. Most isolates were penicillin-resistant (87.5%; 35/40), followed by erythromycin (25.7%; 09/40), cefoxitin (22.8%; 08/40), and ciprofloxacin (05.7%; 02/40). Intermediate resistance was found in tetracycline and chloramphenicol, and all strains remained sensitive to sulfamethoxazole-trimethoprim and gentamycin (Figure 3). No multidrug-resistant bacteria were isolated.

Regarding the study zone, all isolates from areas five and six were resistant to at least one agent, followed by area one (77.7%; 14/18), area four (75.0%; 03/04), area three (66.7%; 02/03), and area two (57.1%; 20/35) (Table 2, Figure 3). It is worth mentioning that most penicillin- and erythromycin-resistant isolates were detected in *Sturnira lilium* (Table 2).

Eight isolates were characterized as cefoxitin-resistant by disc-diffusion, but the *mecA* gene was detected in one isolate (*S. haemolyticus*) from area two (Table 3, Figure 3). The *mecC* gene was not detected. Regarding the penicillin-resistant strains, the *blaZ* gene was detected in 22.9% (8/35) isolates (Figure 3), particularly in areas five (100%; 2/2) and six (100%; 1/1), followed by area one (10.0%; 1/10), area four (33.3%; 1/3) and area two (29.4%; 4/17) (Table 3). Considering the clinically relevant *S. aureus*, all isolates (N = 7) showed resistance to penicillin in the phenotypic test and had the *blaZ* gene detected by PCR, which did not differ among sites or time of collection.

Finally, no association was detected between the presence of resistant isolates and the collection area (*P* = 0.63) or moment of collection (*P* = 0.74).

## 3. Discussion

Wildlife animals are relevant carriers of antimicrobial-resistant bacteria worldwide [33]. Bats belong to a particular group of species that may be potential carriers of resistant bacteria due to their synanthropization and contact with contaminated environments and other species. Their potential to fly to different regions makes them potential disseminators of these bacteria.

In this present research, antimicrobial-resistant *Staphylococcaceae* were isolated from the bats’ skin, which makes them important from the One Health perspective. These bacteria can easily be transferred among individuals in the same colony, to other species or even through occupational transmission. Professionals working with bats are at risk since vampire bat culling is recommended to control cattle rabies. Although bats are handled with proper gloves, this equipment may act as fomites and transfer bacteria to professionals. Feeding and roost habits may favor the spread of resistant microorganisms. For example, the vampire bat (*Desmodus rotundus*) feeds exclusively on blood, particularly from domestic animals, although they can feed on wildlife and humans if needed [34]. An antimicrobial-resistant bacteria on the skin may be transferred to the food source by contact, which can be spread to other animals, humans, or the environment through their daily activities. Bat species, including *Platyrrhinus* spp., *Artibeus* spp., *Pygoderma* spp., and *Sturnira* spp., prefer roosting in trees where several other individuals are. This particular habit favors the microorganism exchange through close contact. The same situation involves the transfer of these bacteria to the trees, which may play a role as carriers and transferors to humans and other animals, mainly when roosts are found in the cities [35].

Wild small mammals (mice, voles, and shrews) were recently characterized as sentinels for the environmental transmission of antimicrobial resistance [36] by detecting more resistant *E. coli* strains in coastal animals than inland ones, in addition to human-associated pathogenic strains. The authors also highlighted concerns about the overall presence of antimicrobial resistance regarding the interaction between animals, the environment, and humans. In this regard, most bat species are long-lived and gregarious, and migration is present in some species [37]. Facing these habits and the consequent close contact with other animals and the environment, bats may also play a role as sentinels for antimicrobial resistance. 

Bats were mostly captured in rural and forested areas, which may explain why most *Staphylococcaceae* isolates were obtained there. However, both sites present low anthropization, which increases the sources of *Staphylococcaceae*. The genus *Staphylococcus* is the most clinically relevant and comprises ubiquitous species that can be found in animals [38], soil [39], water [40], and air [41].

The seasonal activity of bats is usually higher in summer because it is the period with the increased food supply; therefore, it is the right time for pregnant and lactating females to forage [42,43]. Thus, higher activity implies increased contact with other animals and plants, which may explain why most *Staphylococcaceae* isolates were obtained in summer.

The *Staphylococcus* genus comprises 53 recognized pathogenic and commensal species [44] that can be isolated from animals and the environment. *M. sciuri* was the most frequent species isolated in this research, which agrees with the fact that *M. sciuri* is mainly found in animals [45]. However, its importance in human medicine is increasing due to its association with wound infections [46], peritonitis [47], and urinary tract infections [48]. *S. aureus* is a significant pathogen that causes several infections in animals and humans, and it is worth mentioning that these clinically significant species were also isolated from bats. 

It is worth mentioning that *Staphylococcaceae* was mainly isolated from young bats, which may be related to the colonization in females. Similar behavior is observed in bat flies as females usually have a high degree of infestation and transfer the parasites to their offspring [49].

Regarding the genus *Staphylococcus*, few studies have evaluated their colonization and diversity in bat skin. Swab samples were taken from the wing membranes of healthy Livingstone’s fruit bats (*Pteropus livingstonii*) from the Jersey Zoo, United Kingdom [32], and the species were similar to those isolated in the present study. Still, in terms of frequency, *P. livingstonii* mainly were colonized by *S. xylosus,* followed by *S. saprophyticus, S. nepalensis,* and *S. aureus,* differing from the present study [32]. *S. aureus* was also isolated from the skin wounds of bats in Berlin [31]. 

Antimicrobial resistance is a global concern that may cause about 10 million deaths by 2050 [21]. Therefore, studies on the frequency of antimicrobial-resistant bacteria are encouraged not only in humans but in the environment and animals. In addition to deciphering the *Staphylococcaceae* species in bat skin, antimicrobial susceptibility studies are essential for determining their role as carriers of resistant bacteria. 

In the present study, most isolates were resistant to penicillin, a beta-lactam agent extensively used in methicillin-susceptible staphylococcal infections [50]. Since penicillin was massively used after its discovery, *Staphylococcaceae,* particularly the genus *Staphylococcus* and the *M. sciuri,* can become resistant by two central mechanisms: the primary system is the production of a penicillinase encoded by the *blaZ* gene [51], and the second is the production of a penicillin-binding protein (PBP2), which has a low affinity for beta-lactams, encoded by *mecA* and *mecC* genes [51]. 

Although 87.5% (35/40) of isolates were penicillin-resistant, the *blaZ* and *mecA* genes were detected in 22.9% (8/35) and 2.8% (1/35) isolates, respectively. Comparisons between phenotypic detection of penicillin resistance and *blaZ* detection by PCR showed low accuracy [52] and low sensitivity [53], which explains our molecular results. Similarly, Mohammadtaheri et al. (2010) [54] had low sensitivity (63%) when comparing the disc-diffusion method and *mecA* gene detection by conventional PCR.

In research conducted in Chile, Sacristán et al. (2020) [55] collected fecal samples of wild guignas (*Leopardus guigna*) and observed an association between anthropization degree and antimicrobial resistance genes’ prevalence in wildlife. In the present study, however, the bacteria isolated from bat skin was not a good indicator of wildlife and/or environmental exposure to human activity (“zone”; *P* > 0.05). 

The skin *Staphylocociccaceae* of the bats sampled here did not exhibit a multidrug-resistance profile, which differs from feces studies [24]. Fecal samples usually show more diverse microbiota than the skin, mainly due to the ingestion of microbes from food and water. This ingestion reflects in the gut colonization with resistant bacteria that can transfer mobile genetic elements containing resistance genes to non-resistant bacteria. In addition, food and water may have antibiotic residues, increasing the selective pressure in the gut. Therefore, we hypothesize that sample type may impact antimicrobial resistance studies. A cross-sectional study of dairy cattle metagenomes of heavy metal-contaminated and non-contaminated areas was conducted by Gaeta et al. (2020) [56]. The resistome of rumen liquid, deep nasopharyngeal swabs, and fecal samples were compared, and the results showed that fecal samples had the highest frequency of antimicrobial resistance genes. Finally, the possibly less selective pressure on the skin microbiota (compared with other anatomical sites, such as the gut) may also explain the absence of multidrug-resistant bacteria in the captured bats.

Antimicrobial resistance is a One Health issue [22] studied in humans, pets, and farm animals but not enough in wild animals, such as bats. Brazil was the third largest consumer of antimicrobials in livestock in 2010, and it is also projected to be the second country to increase antimicrobial consumption between 2010 and 2030 [57]. These antimicrobials may be dumped in water courses, soil, and plants and reach bats, and this behavior may favor the spread of antimicrobial-resistant bacteria. In addition, bacteria from bats could acquire resistance genes from isolates of humans and other animals through their intense microbial exchange [58].

Although no multidrug-resistant bacteria were detected, the presence of isolates resistant to clinically important drugs warns the need to conduct epidemiologic studies to understand the role of bats as carriers of these bacteria.

## 4. Material and Methods

### 4.1. Bat Capture and Sample Collection

In previous more extensive research, captures were performed from June 2021–April 2022, every three months (one capture in each season of the year), in the optimal conditions for capture, such as new moon, less light, and no rain (when possible). For this study, samples were obtained during summer and autumn (“moment of collection” variable).

Bats were captured in six different areas, according to an increasing order of anthropization (“zone” variable): Area 1 (Forested): a private reserve of environmental preservation of 10,000 acres composed of a dense mountain rainforest and altitude of around 900 m; Area 2 (Rural): a 100 hectares portion of secondary ombrophilous forest with the presence of exotic trees and ornamental species and an average altitude of 800 m; Area 3 (Residential-A): space with several fruit trees, including papaya, banana, an avocado, and a fig; Area 4 (Slum-up to two floors): a small fragment of secondary Atlantic forest, with several *Piper* sp. and eucalyptus trees near to the area; Area 5 (Residential-B—condo buildings): a 1,800 m^2^ condominium building and garage with a vegetable garden and orchard located next to a highway, with several fruit trees including avocados, bananas and mulberries; Area 6 (Industrial): area from the University of São Paulo close to a nearby industrial area, with some non-fruit trees.

Bats were captured using mist nets, which were verified every 30 min. The networks were opened at dusk and closed after 4–5 h [59]. Animals were kept in cloth bags and handled calmly, one by one, by the trained participants. 

Morphometric measurements were estimated for each bat, such as forearm length, sex, age estimate, and weighing using cloth bags and precision spring scales. Then, sterile swabs were rubbed on 147 bats’ wings and backs and immediately stored in ice. Samples were transported to the laboratory, placed in tubes with Peptone Water Buffered (Difco, BD) and incubated at 35 °C ± 2 °C for 12 h.

### 4.2. Microbiologic Tests and Antimicrobial Susceptibility

Ten microliters of each broth with bacterial growth were streaked in Salt Mannitol Agar plates (Difco, BD) following incubation at 35 ± 2 °C for 24 h. Then, bacterial strains were selected and identified using matrix-assisted laser desorption ionization-time of flight mass spectrometry (MALDI-TOF MS).

The antimicrobial susceptibility profile was determined by the disc-diffusion test (Kirby–Bauer method) according to the Clinical Laboratory Standard Institute [60]. Each isolate was streaked in Mueller Hinton agar plates (Difco, BD) and incubated at 35 ± 2 °C for 18 h. Then three to five colonies from the pure cultures were added to a sterile saline buffer, and the suspension was adjusted to the 0.5 McFarland turbidity scale. In three directions, the suspension was streaked onto a Muller Hinton agar plate (Difco, BD). After distributing all antibiotic discs, the plates were incubated at 36 ± 1 °C for 18–24 h. The diameter of inhibition zones around the discs was measured, and the strains were classified as sensitive, resistant or intermediate against each antimicrobial tested according to the CLSI M100 guidelines [60]. The following drugs with respective concentrations (DME, Brazil) were tested: sulfamethoxazole-trimethoprim (SUT 25), gentamycin (GEN 10), chloramphenicol (CLO 30), tetracycline (TET 30), cefoxitin (CFO 30), penicillin (PEN 10), erythromycin (ERI 15), and ciprofloxacin (CIP 05). The *Staphylococcus aureus* ATCC 25923 was used as a negative (sensitive) control. A laboratory strain (*Staphylococcus epidermidis* strain USP-LZB-G06) [61] was used as a positive control for resistance to CFO, PEN, CIP, SUT, and ERI.

### 4.3. DNA Extraction 

DNA was obtained using the protocol described by Fan et al. (1995) [62]. Pure colonies were added to 50 microliters of phosphate buffer saline 1X (2.6 mM NaH2PO4, 7.4 mM NaHPO_4_, 10 mM NaCl, pH 7.2), boiled at 100 °C for 10 min, and put on ice for five minutes. At the end of the previous step, microtubes were centrifuged at 14,000× *g* rpm for 10 min, and the supernatant was stored in new sterile microtubes at −20 °C until processing. 

### 4.4. Spa Typing

Spa Typing analysis was performed for all *S. aureus* isolates. The protein A region was amplified by PCR, according to Harmsen et al. (2003) [63]. Briefly, the polymerase chain reaction (PCR) was performed in a total volume of 25 µL consisting of: 12.5 µL of 2× GoTaq Green Master Mix (Promega), 10 µM of each primer, and 9 µL of nuclease-free water. The amplification cycle was the first denaturation at 95 °C for five minutes, 35 cycles of denaturation at 92 °C for 45 s, annealing at 60 °C for 45 s, extension at 72 °C for 90 s, and a final extension step at 72 °C for 10 min. Ten microliters of each amplicon were purified using four microliters of ExoProStar (Sigma-Aldrich, St. Louis, MO, USA) followed by incubation (37 °C for 15 min and 80 °C for 15 min, respectively). Then, a mixture containing one microliter of BigDye (Thermo Fisher™), 0.5 µL of each primer, and 1.5 µL of buffer 5× (included in the BigDye Terminator v3.1 cycle sequencing kit). Three microliters of this mixture were distributed in a 96-deep well plate, followed by seven microliters of the purified DNA. After centrifugation at 1000 rcf for 60 s, the plate was incubated in a thermocycler: one cycle at 96 °C for 60 s, 60 cycles of 96 °C for 10 s, 50 °C for five seconds and 60 °C for four minutes. Then, 2.5 µL of EDTA 125 mM and 30 µL of 100% ethanol were added to each well for DNA precipitation. After drying at room temperature, the plate was centrifuged at 2250 rcf for 30 min at 15 °C, and 30 µL of 70% ethanol was added to each well. The plate was centrifuged at 1650 rcf for 15 min at 15 °C. Then, the plate was incubated at 95 °C for 10 min with the lid up. Finally, 10 µL of Hi-Di™ was added to each well, and the plate was placed in the Eppendorf sequencer for sequencing by the Sanger method. The obtained files were uploaded to Bionumerics (bioMérieux, France) for complete characterization of the Spa Type. 

### 4.5. Amplification of Resistance Genes

A polymerase chain reaction was performed to detect the *blaZ*, *mecA* and *mecC* genes. The *blaZ* gene was amplified according to Martineau et al. (2000) [64]. Briefly, the PCR was performed in a total volume of 25 µL consisting of: 12.5 µL of 2× GoTaq Green Master Mix (Promega), 75 µM of each primer, and 9.5 µL of nuclease-free water. The amplification cycle was the first denaturation at 94 °C for five minutes, 30 cycles of denaturation at 94 °C for 20 s, annealing at 60 °C for 30 s, and extension at 72 °C for 90 s, and a final extension step at 72 °C for five minutes. The *mecA gene* amplification was performed according to Mehrotra et al. (2000) [65]. The PCR was performed in a total volume of 15 µL consisting of 7 µL of 2× GoTaq Green Master Mix (Promega), 10 µM of each primer, and 5.4 µL of nuclease-free water. The amplification cycle was the first denaturation at 94 °C for five minutes, 34 cycles of denaturation at 94 °C for two minutes, annealing at 52 °C for two minutes, extension at 72 °C for 60 s, and a final extension step at 72 °C for seven minutes. Finally, the *mecC* gene was amplified according to Paterson et al. (2012) [66]. The PCR was performed in a total volume of 15 µL consisting of 7 µL of 2× GoTaq Green Master Mix (Promega), 10 µM of each primer, and 5.4 µL of nuclease-free water. The amplification cycle was the first denaturation at 94 °C for five minutes, 34 cycles of denaturation at 94 °C for 30 s, annealing at 52 °C for 30 s, extension at 72 °C for 60 s, and a final extension step at 72 °C for 10 min. The amplicons were observed by ultraviolet light after 40-min gel electrophoresis for 35 min. Laboratory isolates were used as positive controls. 

### 4.6. Data Analysis

Results were shown as absolute and relative frequencies. Comparisons between the number of isolates and variables were performed using Pearson’s Chi-Square test with continuity correction, except for “collection area”, when the Kruskal–Wallis test was used. The association between “resistance”, “collection areas”, and “sampling moment” was evaluated using the Fisher exact test. Diversity analysis was performed using the “vegan” package from RStudio [67], and diversity indexes were compared using the Kruskal–Wallis test. Figures were elaborated using the “ggplot2” package [68] from RStudio [67] and R v.4.0.5. Variables were considered significant when *P <* 0.05.

## Figures and Tables

**Figure 1 antibiotics-12-00331-f001:**
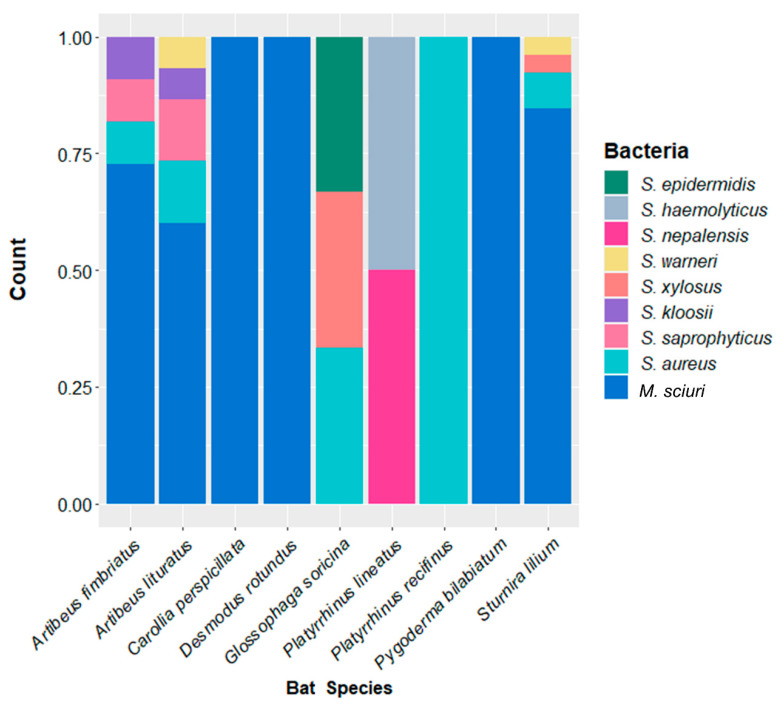
Stacked bar plot showing the relative abundance of *Staphylococcaceae* isolated from the skin of each bat species.

**Figure 2 antibiotics-12-00331-f002:**
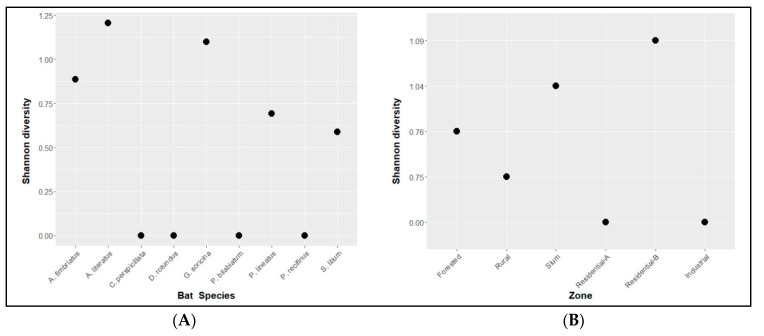
Relationship between Shannon diversity index for *Staphylococcaceae* isolated from skin swab samples and bat species (**A**; *P* = 0.43) and zone of collection (**B**; *P* = 0.41). Bat species are distributed alphabetically, and zones are distributed from the lowest to the highest degree of anthropization.

**Figure 3 antibiotics-12-00331-f003:**
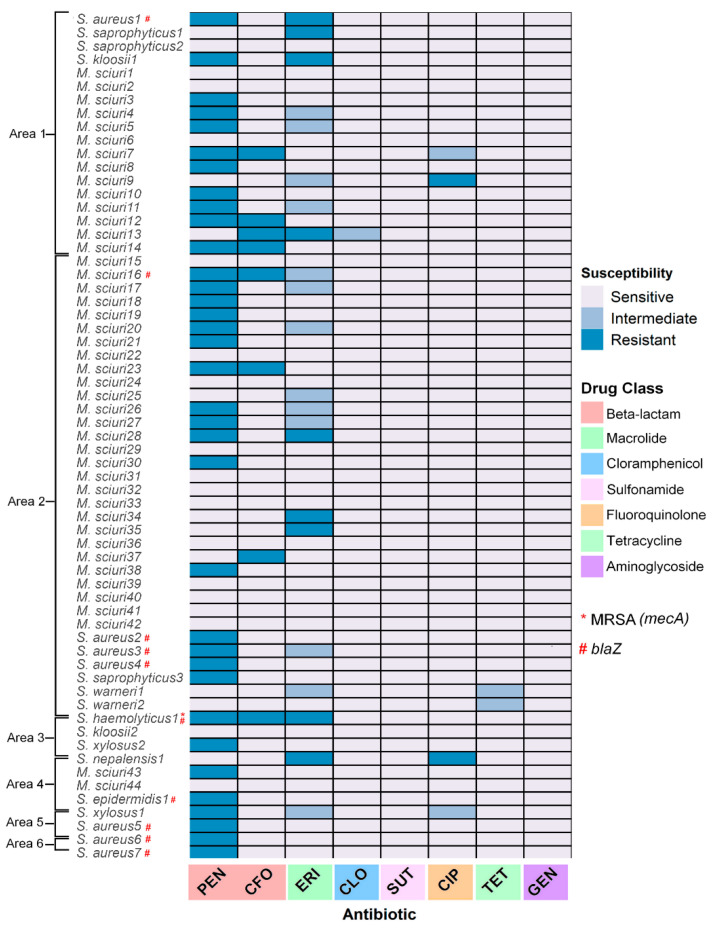
Antimicrobial susceptibility profile of 63 *Staphylococcaceae* strains from skin swab samples from bats, distributed according to study zone and antibiotic classes. Zones are arranged from the lowest to the highest degree of anthropization.

**Table 1 antibiotics-12-00331-t001:** Frequency of the ten *Staphylococcaceae* species isolated from skin swab samples according to bat species count (BSC). Results are arranged in descending order of the total number of isolates.

Bat Species	*Staphylococcus* spp.	% (N/BSC)	Total Isolates % (N/BSC)
*Desmodus rotundus*	*M. sciuri*	100 (01/01)	100 (01/01)
*Pygoderma bilabiatum*	*M. sciuri*	100 (01/01)	100 (01/01)
*Carollia perspicillata*	*M. sciuri*	75.0 (03/04)	75.0 (03/04)
*Platyrrhinus recifinus*	*S. aureus*	50.0 (01/02)	50.0 (01/02)
*Platyrrhinus lineatus*	*S. nepalensis*	25.0 (01/04)	50.0 (02/04)
	*S. haemolyticus*	25.0 (01/04)	
*Sturnira lilium*	*M. sciuri*	39.3 (22/56)	46.4 (26/56)
	*S. aureus*	3.6 (02/56)	
	*S. warneri*	1.8 (01/56)	
	*S. xylosus*	1.8 (01/56)	
*Artibeus lituratus*	*M. sciuri*	23.7 (09/38)	39.5 (15/38)
	*S. aureus*	5.3 (02/38)	
	*S. saprophyticus*	5.3 (02/38)	
	*S. warneri*	2.6 (01/38)	
	*S. kloosii*	2.6 (01/38)	
*Artibeus fimbriatus*	*M. sciuri*	28.6 (08/28)	39.3 (11/28)
	*S. aureus*	3.6 (01/28)	
	*S. saprophyticus*	3.6 (01/28)	
	*S. kloosii*	3.6 (01/28)	
*Glossophaga soricina*	*S. epidermidis*	9.1 (01/11)	27.3 (03/11)
	*S. xylosus*	9.1 (01/11)	
	*S. aureus*	9.1 (01/11)	

N = number of isolates.

**Table 2 antibiotics-12-00331-t002:** Frequency of resistant *Staphylococcaceae* strains to cefoxitin, ciprofloxacin, penicillin, and erythromycin, and detection of *mecA* and *blaZ* genes according to the study zone. Zones are arranged from the lowest to the highest degree of anthropization.

Zone	Bat Species	Total Resistant Isolates % (N/T) *	Cefoxitin	Ciprofloxacin	Penicillin	Erythromycin	*mecA*	*blaz*
Forested	*Pygoderma bilabiatum*	100 (1/1)	0	0	1	0	0	0
	*Carollia perspicillata*	100 (2/2)	1	0	2	0	0	0
	*Sturnira lilium*	83.3 (5/6)	2	0	4	2	0	1
	*Artibeus lituratus*	75.0 (6/8)	1	1	4	2	1	0
	Total Area 1	77.8 (14/18)	4	1	10	4	1	1
Rural	*Platyrrhinus lineatus*	100 (1/1)	1	0	1	1	1	1
	*Platyrrhinus recifinus*	100 (1/1)	0	0	1	0	0	1
	*Sturnira lilium*	66.7 (12/18)	1	0	9	3	1	2
	*Artibeus fimbriatus*	44.4 (4/9)	1	0	4	0	1	1
	*Artibeus lituratus*	50.0 (2/4)	1	0	2	0	1	0
	Total Area 2	57.1 (20/35)	4	0	17	4	4	5
Residential-A	*Platyrrhinus lineatus*	100 (1/1)	0	0	1	0	0	0
	*Glossophaga soricina*	100 (1/1)	0	1	0	1	0	0
	Total Area 3	66.7 (2/3)	0	1	1	1	0	0
Slum	*Artibeus lituratus*	100 (1/1)	0	0	1	0	0	0
	*Glossophaga soricina*	100 (1/1)	0	0	1	0	0	1
	*Sturnira lilium*	50.0 (1/2)	0	0	1	0	0	0
	Total Area 4	75.0 (3/4)	0	0	3	0	0	1
Residential-B	*Artibeus lituratus*	100 (1/1)	0	0	0	1	0	0
	Total Area 5	100 (1/1)	0	0	0	1	0	0
Industrial	*Glossophaga soricina*	100 (1/1)	0	0	1	0	0	1
	*Artibeus lituratus*	100 (1/1)	0	0	1	0	0	1
	Total Area 6	100 (2/2)	0	0	2	0	0	2

* Refers to isolates resistant to at least one antibiotic agent. N = number of isolates. T = total of isolates.

**Table 3 antibiotics-12-00331-t003:** Distribution of cefoxitin (CFO)-resistant/methicillin-resistant *Staphylococcaceae* (MRS), penicillin (PEN)-resistant isolates and detection of *mecA* and *blaZ* genes according to study zones. Zones are arranged from the lowest to the highest degree of anthropization.

Zone	Species (N)	CFO-Resistant/MRS	*mecA*	PEN-Resistant	*blaZ*
Forested	*S. aureus* (3)	0	-	100.0 (3/3)	100.0 (3/3)
	*S. haemolyticus* (1)	100.0 (1/1)	100.0 (1/1)	100.0 (1/1)	100.0 (1/1)
	*S. saprophyticus* (1)	0	-	100.0 (1/1)	0
	*M. sciuri* (28)	10.7 (3/28)	0	7.1 (2/28)	0
	*S. warneri* (2)	0	-	0	-
	***TOTAL* (35)**	**11.4 (4/35)**	**2.9 (1/35)**	**20.0 (7/35)**	**11.4 (4/35)**
Rural	*S. aureus* (1)	0	-	100.0 (1/1)	100.0 (1/1)
	*S. kloosii* (1)	0	-	100.0 (1/1)	0
	*S. saprophyticus* (2)	0	-	0	-
	*M. sciuri* (14)	28.6 (4/14)	0	64.3 (9/14)	0
	***TOTAL* (18)**	**22.2 (4/18)**	**0**	**61.1 (11/18)**	**5.6 (1/18)**
Residential-A	*S. kloosii* (1)	0	-	0	-
	*S. nepalensis* (1)	0	-	0	-
	*S. xylosus* (1)	0	-	100.0 (1/1)	0
	***TOTAL* (3)**	**0**	**-**	**33.3 (1/3)**	**0**
Slum	*S. epidermidis* (1)	0	-	100.0 (1/1)	100.0 (1/1)
	*M. sciuri* (2)	0	-	50.0 (1/2)	0
	*S. xylosus* (1)	0	-	100.0 (1/1)	0
	***TOTAL* (4)**	**0**	**-**	**75.0 (3/4)**	**25.0 (1/4)**
Residential-B	*S. aureus* (1)	0	-	100.0 (1/1)	100.0 (1/1)
Industrial	*S. aureus* (2)	0	-	100.0 (2/2)	100(2/2)

N = number of isolates.

## Data Availability

All data are available upon request.

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
