# Peer review of "Bats Are Carriers of Antimicrobial-Resistant Staphylococcaceae in Their Skin"

_antibiotics, 2023, doi:10.3390/antibiotics12020331_

Round 1

Reviewer 1 Report

The study covers a relevant topic and is well-written in most parts. The work is interesting. I have not any comments. I confirm the acceptance of this paper.

Author Response

The authors thank to the Editor and reviewer 1 for the thoughtful and thorough review. 

1) The study covers a relevant topic and is well-written in most parts. The work is interesting. I have not any comments. I confirm the acceptance of this paper.

A: We would like to thank the reviewer for this comment and consider our manuscript for publication in Antibiotics.

Reviewer 2 Report

This is a report highlighting the presence of antibacterial resistant Staphylococcus spp. on bat skin. The data presented here are very interesting, yet the significance of the findings is rather lost in the discussion.

Importantly, the material and methods section is missing information regarding the detection of blaZ and mecA genes by PCR. Section 4.2 (L267), needs to include far more detailed descriptions of the analytical procedures, similarly to the first section (L241) which is very detailed indeed. In fact, the capture and collection procedure is very well described so much so that it takes away from the analytical procedures, which is not suitable entirely for the SI this manuscript is being submitted to. So please do try to adjust the level of detail in the description of the analytical procedures. 

The discussion does not provide clear links between the significance of the findings described in the manuscript to the One Health concept (which is mentioned by the authors), nor to intensifying action plans to control the spread of resistant bacteria (L28-29). This is rather unfortunate. I also do not agree with the statement in L12 that bats have only recently been acknowledged as potential reservoirs of zoonotic viruses and bacteria--bats are in fact known as reservoirs of zoonotic viruses and bacteria and this knowledge dates back many years. Please try to make your findings in the discussion section more relevant to the One Health concept by possibly in fact highlighting the potential of bats as more than just reservoirs but also as sentinels for antimicrobial resistance in the environment. In fact, as it stands now (L156-160) the need to intensify efforts to control the spread of resistant bacteria is not very clear in relation to the data obtained here. It is suggested here that bats may be relevant for human infection only (or most significantly) in the case of actual handling of the bats by professionals (L160), which is certainly true but doesn't actually justify the need for control of spread of resistant bacteria--there are far easier methods to achieve protection for bat handlers.  

Round 2

Reviewer 3 Report

Publication of the paper in its present form is recommended. The only suggestion: instead of M. sciuri Mammaliicoccus sciuri should be written in line 23